# Evidence of Differences and Discrimination in the Delivery of Care: Colorectal Screening in Healthy People and in the Care and Surveillance of Patients with Inflammatory Bowel Disease

**Afffa Farrukh [1],[†] and John F. Mayberry [2],[*],[†]**

[1]  Digestive Diseases Centre, University Hospitals of Leicester, Gwendolen Road, Leicester LE5 4PW, UK; farrukh_affi@yahoo.com

[2]  Department of Digestive Diseases, Leicester General Hospital, Gwendolen Road, Leicester LE5 4PW, UK

[*]  Correspondence: johnfmayberry@yahoo.co.uk; Tel.: +44-1162-584-787

[†]  These authors contributed equally to this work.

**Abstract:** Objectives: In this review the management of colorectal disease will be investigated as an exemplar of common practice in the UK in an attempt to identify factors responsible for the more general experiences of patients from ethnic minorities. Within this field such populations have a lower uptake of cancer screening programmes and their experience of day-to-day care for chronic gastrointestinal disorders is poor. Study design: PubMed and Google Scholar were reviewed in 2016 to identify publications concerning colorectal screening in patients with inflammatory bowel disease and healthy communities. Methods: Data were extracted from each paper and the references exploded to identify other potential reports. Results: It is reported that barriers exist both at individual and access levels but little has been done to overcome these. There have been a number of suggestions as to how to provide equitable access, but there is a clear need to ensure that these are evidence based and have been tested and shown to be effective in clinical trials. Conclusions: Clearly, current systems of surveillance and screening will only make a difference if they provide effective and acceptable services to all potential clients. Most programmes fail to address the specific risks and anxieties of minority groups, which are thought to be poorly compliant. This review considers those factors that may play a part and suggests approaches that could overcome these deficiencies. Some clues as to these factors may come from work with patients with chronic disorders.

**Keywords:** surveillance; screening; colon; inflammatory bowel disease; ethnicity; chronic disease

## 1. Introduction

The delivery of care should be equitable and free of discrimination. However, ethnic minorities still experience institutional racism with evidence of a range of disparities in the delivery of healthcare across many specialties. In the UK, such differences are currently being reported in mental health [1], diabetes and coronary artery disease [2], oesophageal and gastric cancers [3], as well as hip and knee replacement [4]. In this review the management of colorectal disease will be investigated as an exemplar of common practice in an attempt to identify factors responsible for such experiences. Within this field ethnic minority populations have a lower uptake of cancer screening programmes in the UK [5–7]. It is reported that barriers exist both at individual and access levels but little has been done to overcome these. Sociodemographic factors cannot fully explain the suboptimal participation by ethnic minority groups and further research is essential, especially as some dispute the existence of such disparities in care. This is despite the findings of studies, as in Leeds, where nurses were found to

have limited understanding of the local Pakistani community and were ill-prepared to deal with their needs with associated evidence of racist attitudes [8].

Disparities in healthcare access affecting ethnic minority and "hard to reach" groups present a complex challenge to clinicians, managers and policy-makers. There have been suggestions as to how to provide equitable access, but there is a clear need to ensure that these are evidence based and have been tested and shown to be effective in clinical trials [9,10].

There is a growing movement across the world to introduce screening programs for the early detection of colon cancer or precancerous states, but unfortunately the literature on various cancer screening programmes persistently shows lower uptake especially by ethnic minorities [11–25]. A study from almost 20 years ago showed that the incidence of colorectal cancer amongst south Asian migrants to the UK was increasing [26]. However, little has been done in spite of this constant and consistent message to improve the situation. Published research on interventions to improve screening uptake in such groups is rare and suggests that the demand for effective approaches to rectify these barriers to cancer screening programmes needs further evaluation. In particular, most programmes fail to address the specific risks and anxieties of minority groups, which are thought to be poorly compliant. However, studies from China [27] and India [28] have shown the indigenous populations to be at comparable risk of cancer in ulcerative colitis and there are no reasons to believe that the situation would be any different for migrant communities from these areas. Despite these observations there has been little work on the impact of diversity on the care of patients with polyposis syndromes and long-term ulcerative colitis, and there is little evidence that funding bodies have plans to encourage research into interventions which would change this situation [29,30]. In these conditions, which are known to predispose to colorectal cancer, do patients have equal access to surveillance and so benefit from early detection of dysplasia or cancer? In the case of ulcerative colitis, can quality of care be assessed by examining whether people from diverse backgrounds experience similar approaches to monitoring the effect of aminosalicylate (5ASA) compounds on renal function or have comparable access to expensive treatments such as biologics and ileoanal anastomotic surgery? Such questions should underpin all aspects of research on quality of care and, at a time when we are all encouraged to be involved in "Quality Improvement Projects", this is an area of obvious need.

## 2. Review of Published Work

*Study Design*

PubMed and Google Scholar were reviewed in 2016 to identify publications concerning colorectal screening in patients with inflammatory bowel disease and healthy communities.

Review

There is recent evidence that patients from ethnic minority groups demonstrate significant differences in access to cancer screening as well as uptake of the services. Breast and cervical cancer screening are well established in the UK, but figures consistently show a lower uptake by ethnic minority groups, especially South Asians. Although uptake in the Afro-Caribbean population has been reported to be equal to the white population, little is known about communities such as the Chinese [9,10]. In a MORI survey in primary care in 1995, Carr-Hill and Rudat [9] reported a relatively high uptake of cervical screening by Afro-Caribbean people (87%), but a much lower uptake by South Asian women (70% Indian; 54% Pakistani; and 40% Bangladeshi). However, there are aspects which limit the applicability of these findings to colorectal cancer (CRC) screening. Firstly, breast and cervical screening programmes in the UK differ in organisation and delivery. Secondly, both programmes exclude men and so provide no evidence on how men would respond to CRC screening. Work in the UK on mass screening of healthy people by flexible sigmoidoscopy found that although intentions were comparable across various cultural groups, attendance was considerably lower among Asians (54%) compared with White (69%) or Black (80%) respondents [6]. International literature, mainly

from USA, does provide evidence of lower uptakes of faecal occult blood testing (FOBT), flexible sigmoidoscopy and colonoscopy amongst minority ethnic populations [11,12]. The UK CRC screening pilot programme amongst the general population reported an overall uptake of 62% for faecal occult blood testing (FOBT) for English people. This was considered acceptable, but uptake was poor in the South Asian community, with figures for Muslims as low as 32%. The highest figure amongst South Asians was for Hindus, but was still only 44%. Unfortunately, analysis of FOBT uptake rates in this study was dependent on surname recognition software and so was unable to distinguish Afro-Caribbeans from English people [5]. In a subsequent analysis of the first 2.6 million people to take part in the national screening program in the UK, the most ethnically diverse areas had the lowest uptake (38%), compared to other communities where the figure was between 52 and 58%. These differences were independent of social, economic and personal status. However, ethnic disparities were more pronounced in men but equivalent across age groups [6,7].

Where studies have examined sociodemographic factors [5,9], it is commonly reported that these cannot fully explain such variations. In the general population the uptake of FOBT screening remains two and a half times lower amongst Muslims and Sikhs and about twice as low amongst Hindus. Colonoscopy uptake rates were also significantly lower amongst Asians (55%) compared with 74% for non-Asians [5]. However, there has been no work on uptake rates amongst patients with chronic gastrointestinal diseases, such as ulcerative colitis. Indeed there has been little work on any aspect of healthcare amongst this group of patients and the impact of diversity issues, such as age, gender, ethnicity, ability and sexuality. In the one report on long-term care of patients with ulcerative colitis, patients of South Asian origin clearly had a poorer experience than English patients in the same community. For example, although the difference did not reach statistical significance, only 32% of South Asian patients underwent surveillance colonoscopy compared to 43% of English patients.

A study from the USA suggested that low uptake of screening for colorectal cancer in the general population may be due to a lack of awareness and inadequate provider counselling rather than a poor acceptance of the scheme [11–13]. To encourage participation and improve compliance, physician recommendation was suggested to have a major impact [14–18]. The problems generated by developing schemes with an English language basis and failing to address cultural issues are well established in many studies. Undoubtedly such an approach is likely to strengthen perceived discrimination, as has been demonstrated amongst American Muslim women in breast cancer screening programs. Language issues are clearly of major significance. An American study has shown that bilingual immigrants are better able to negotiate the healthcare system than those fluent in English or their native language alone [20]. African migrants in Guangzhou, China found the language barrier to effective healthcare a major issue and this led to a distortion in the patient–doctor relationship. Doctors were seen to be too hurried and to lack cultural awareness [21]. Such findings are important because it is generally agreed that poorly educated people are less likely to receive counselling about CRC screening from their health provider. Individual knowledge of colon cancer, lack of perception of illness in the absence of symptoms [19,22,31], health literacy [22–25] and fear of further investigations also deter participation. Lack of awareness is globally accepted as a proxy for lower compliance. However, a study from Leicester on CRC screening in a workplace setting reported that the lowest uptake of FOBT was amongst doctors and managers (26%) when compared with clinical support staff (56%). This difference was not further explored, but emphasises the fact that knowledge of CRC, its early detection, prevention and management are not necessarily linked to high uptake rates in screening programs [31].

In polyposis syndromes there is evidence that the stigma associated with cancer or family risk play an important role in preventing South Asian people seeking genetic counselling and associated preventive screening [29]. Significant differences regarding attitude towards disease and its impact on daily life are also found amongst ethnic groups with inflammatory bowel disease (IBD) [30]. We know that, in general, adherence to screening programs amongst people with ulcerative colitis is low (Table 1), but little is known of the reasons for this noncompliance [32–35]. In the case of patients

from ethnic minorities we have to question the role of senior medical personnel in ensuring they receive appropriate explanations about the disease, appropriate support and community education. Clearly there is evidence that some clinicians shy away from dealing with patients from a different community to their own [36].

**Table 1.** Adherence/participation rates in surveillance programs in ulcerative colitis.

| Country | Study Design | Year | Total Patients | Non Compliant |
|---------|--------------|------|----------------|----------------|
| Sweden [35] | Prospective | 1977–2002 | 211 | 36/211 (17%) |
| Sweden [34] | Prospective | 1977–1991 | 131 | 13/131 (10%) |
| Italy [32] | Prospective | 1980–2000 | 65 | 29/65 (45%) |
| Italy [33] | Prospective | 1989–1992 | 65 | 15/65 (23%) |

There is growing evidence of significant differences in many aspects of the care of patients with gastrointestinal diseases from minority communities. In North Manchester, Redbridge and Leicester the number of South Asian patients who received biologic treatment for Crohn's disease was significantly less than British/White patients [37]. In a recent UK nationwide review of colectomies for ulcerative colitis, Indians had a significantly higher rate than White Europeans (11 vs. 7%. In contrast, Pakistanis had a similar (7.0%) and Bangladeshis a significantly lower (4.7%) colectomy rate [38]. Twenty years ago there was evidence that the disease may have been milder in the South Asian community [39]. In addition there was significant concern about the possibility of a stoma amongst South Asian patients [40]. Although within a generation disease severity was comparable between South Asian and White patients in Leicester [41], the recent study of a higher rate of colectomy amongst Indian patients raises serious questions about their access to effective alternative expensive medical treatments. With the increased availability of biologics for management of the condition, the observation that access to such therapy is not equitable in the UK emphasises the need for clinicians to urgently address these issues. For those who doubt the reality of such differences in care there is added confirmation from the USA. In both Atlanta and Baltimore, White patients were significantly more likely to receive infliximab, while in Miami, the ratio for Hispanics was 22% compared with 56% for White-Americans [42–44].

## 3. Discussion

In the 2011 Census, Black and ethnic minorities constituted almost 11% (6.2 million) of the population in the UK. South Asians accounted for about half this population. There are also 1.15 million Afro-Caribbean people in the UK. These groups are younger than the White population, which means that the ethnic minority percentage of the population is likely to grow in coming years [45]. Following the implementation of the Race Relation Amendment Act 2000, it is a statutory duty on all NHS agencies to "have due regards to the need to eliminate unlawful discrimination" and maintain racial equality. Access to care is a product of the interaction between provider services, users and planners. Quality of care is defined in terms of client centred services, meeting needs and expectations. It is an ongoing process that incorporates clients' rights and their satisfaction with the system. It entails access to accurate, appropriate, understandable and unambiguous information at convenient times with no physical barriers. It helps clients make a well-judged decision based upon information, understanding and options, i.e., informed choice rather than coercion.

Provider performance is the building block of accountability in a system, which is meant to offer a sustainable and continuous flow of services to satisfy its clients which in turn is measured by their level of participation. This collective vision of quality woven together from individual voices from different communities can give systems, such as screening and surveillance, a new life, where individuals are no more just patients, but rather, healthcare consumers. It not only holds clients responsible for their decisions by engaging them in defining and supporting the quality of services they want, but also helps

providers identify and recognise root causes of problems with their services. However, as Diangelo recently pointed out, "if I am not aware of the barriers you face, then I won't see them, much less be motivated to remove them" [46]. There is more needed than simple education. Although education can improve healthcare workers knowledge of minority communities there is no evidence that it will change clinical practice [47]. The work of van Ryn et al. has shown that where clinicians have an awareness of the issue and are prepared and able to address institutional racism in their delivery of care, then such individuals will be able to make changes in their own practices and also that of their institutions [48].

It is clearly necessary to broaden access so as to ensure "equitable" care in this vision of the health industry in UK. Traditionally access has been measured in terms of utilisation and this is based on "Demand" and "Supply" concepts. This system is now complicated by the fact that we can identify those who do or do not use its services but are unable to explain why the benefits experienced by one community are not utilised by others. Though the norms of provider behaviour will determine what is deemed accessible by users, the decision to seek formal care will be influenced by how the services on offer are perceived by potential users. Healthcare is a complex system where every participant interacts with each other in a dynamic way, where changes in one element can alter the context of all others as well as subsequently being influenced by them. It involves individuals making discrete choices based on their own assessment of options. These decisions are often a result of formal or informal negotiations between multiple members of the community, rather than simply an individual weighing up different courses of action. However, there are only limited data on how this plays out in practice for people with chronic gastrointestinal diseases, such as ulcerative colitis. So, if the healthcare system is meant to be client oriented, it has to take into consideration what is important to its consumers so as to ensure the ongoing trust of existing users and their recommendation of it to others. For example a persistently lower uptake of cancer screening by South Asians reflects the fact that cancer screening services in the UK are not sensitive to the needs of ethnic minorities. Barriers exist for both individuals and at access levels including cultural differences, conscious or unconscious racial bias, professional and cultural insensitivities, language and literacy, stress from experiences of discrimination and unmeasured socioeconomic variables. All are intertwined and can create a distrust of healthcare services. Unfortunately little has been done to introduce and evaluate the effectiveness of implementations once such barriers are identified.

The purpose of this review is to expose the existence of barriers to care amongst patients with chronic gastrointestinal disorders such as inflammatory bowel disease and their link to diversity. Its secondary purpose is to assess the magnitude of these disparities. There is a clear need for a more formal investigation through qualitative techniques of the reasons behind these barriers. Such qualitative techniques must involve and, ultimately, be led by truly representative members of those groups whose access to quality care has been impaired. It cannot depend on token members, acceptable to a majority of the healthcare providers, and representative of no one. However, before gastroenterologists can engage with the communities they serve they need, on an individual basis, to assess whether their service truly delivers care regardless of ethnicity, religion, gender, sexuality or ability. For example, how many of us have engaged with the signing community, although there are an estimated 125,000 users of British Sign Language according to a recent general practice poll [49]? Some basic practical tests we can all apply include

1. Does my clinic composition represent the community in which I work?
2. Do I personally ensure that I see an appropriate cross-section and make use of professional translation services?
3. Am I properly and correctly informed about the cultural background of my patients?

As gastroenterologists we need to develop formal links with minority communities, encourage them to set up patient groups and, through formal techniques, such as the Delphi Method, start to

identify ways of removing barriers of access to care and ensure that the care we do deliver is equitable and fit for purpose.

**Author Contributions:** Both authors contributed equally to the collection of data, its interpretation and writing of the review.

**Funding:** This research received no external funding.

**Conflicts of Interest:** The authors have no conflict of interest.

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
