# Peer review of "Evidence of Differences and Discrimination in the Delivery of Care: Colorectal Screening in Healthy People and in the Care and Surveillance of Patients with Inflammatory Bowel Disease"

_gastrointestdisord, doi:10.3390/gidisord1020020_

Round 1

Reviewer 1 Report

This is an important review that underlines the problem of existing barriers to care amongst patients with chronic gastrointestinal disorders such as inflammatory bowel disease. It is not clear if based on available data it will be possible to provide some metrics – epidemiology studies on different populations. Over all if not available this review should be a clear inspiration to do so. The Authors reached they secondary purpose, which was to assess the magnitude of analysed disparities. 

Author Response

We have now added 3 references which have demonstrated that:

Colon cancer is increasing amongst South Asians in the UK

The risk of colon cancer in ulcerative colitis in Indian and Chinese patients in their home countries is directly comparable to the West.

There is therefore no epidemiological reason to indicate a lower risk and for there to be lees need for a high uptake of surveillance

Reviewer 2 Report

In this manuscript, the authors discussed a number of publications to highlight the differences in the delivery of healthcare in colorectal diseases, and attempted to discuss factors responsible for such phenomenon. While the manuscript is successful in raising awareness of healthcare inequality, a few issues need to be addressed:

(1) A major and fatal issue with this manuscript is the lack of an objective tone. The best example is the beginning of the introduction of this manuscript, which adopted the genre of best-seller nonfiction books, stating the stance and values of the authors. This directly calls the scientific objectivity of this manuscript into question.

(2) The manuscript spent a lot of page space talking about the phenomenon of healthcare inequality at the expense of discussing underlying reasons. Insights into the reasons behind the healthcare delivery inequality need to be significantly expanded.

(3) The authors did not do a sufficient job stating the significance of this work, i.e. why is it important to study institutional racism in colorectal diseases as opposed to other types of illnesses. Please provide epidemiological data to support. In other words, the data discussed in this manuscript focused solely on the fact that delivery of care is not distributed equally among ethnic minority groups; data showing its consequences would need to be discussed. 

Along the same line, the entire manuscript jumps back and forth between diseases, making it appear to lack a clear colorectal focus.

(4) Data discussed in the manuscript solely mentioned developed countries, which are far from being the most ethnic diverse countries in the world. A comprehensive review should cover the care delivery distribution from Africa, which has many of the most ethnically diverse countries in the world. Though data from these countries might be lacking, they should be at least mentioned. Health care data from developing countries, such as China and India, which also have many ethnic groups, should be discussed as well.

Author Response

We have removed significant sections from the introduction. 

Insight into the reasons for disparities in healthcare delivery would itself be largely speculative as there have been few studies which have provided explanations or suggestions. We have added a reference on the role of language but not added further thoughts which do not have an evidence base. 

We have added epidemiological references to show that colorectal cancer is increasing amongst Asian patients in the UK and other epidemiological studies which show that cancer risk in ulcerative colitis is the same in China and India as in the West. Hence there is no lesser need for effective screening in the moigrant communities. However, the number of studies whch have looked at these issues is limited

We looked specifically at colorectal disease as there is a major drive throughout the Western World to develop effective screening programs and those programs as documented in the review are not reaching minority communities. We have added a statement to this effect.

Unfortunately in order to gain some understanding of the widespread nature of the problem it is necessary to draw on other disease gropus and this inevitably means that the paper does at times jump from one disease group to another. the purpose of the paper is to try to identify some core issues.

It is true that there are many communities which are very diverse and that many/most are not in the developed world. We repreated our pubmed review and also utilised Google Scholar and widened the search terms. This only identified one paper from such communities and again it deal with a  migrant community, namely Africans in China. There were no papers looking at the provision of disparate care within a developing country to the indigenous populations of that country. the Chinese paper has been added to the text.

7. We have added in health care data from India and China about colorectal cxancer risk in ulcerative colitis and access to healthcare by migrants into China. We have not discuss the nature of healthcare in these countries as there are no published studies on differences in access between different communities.

Round 2

Reviewer 2 Report

The authors made significant improvement on the objective tone of this manuscript. A limited number of studies in developing countries were also included in the discussion. 

Mechanistic insights are still lacking, but the authors seem to have already tapped into the room for improvement.

Author Response

We have added three further references, tracked in blue, which address the issues of what might be done to overcome institutional racism in the delivery of health care. Unfortunately although all papers call for action there is no clear evidence of any effective action that can be taken with the exception of van Ryn's work - which calls for awareness of the issue and individual responses by clinicians